# The Sensory Input from the External Cuneate Nucleus and Central Cervical Nucleus to the Cerebellum Refines Forelimb Movements

**DOI:** 10.3390/cells14080589

**Published:** 2025-04-13

**Authors:** Chidubem Eneanya, George M. Smith

**Affiliations:** Department of Neural Sciences, Lewis Katz School of Medicine, Temple University, 3500 North Broad Street, Philadelphia, PA 19140, USA; dubieneanya@temple.edu

**Keywords:** spinal cord injury, synaptic plasticity, regeneration, motor control, sensory system, proprioception, rehabilitation, functional recovery

## Abstract

Goal-directed reaching movements are extremely accurate to the point that the location, placement, and speed of the limbs are specific from trial to trial. These movements require descending motor commands and feedback modulation from ascending sensory information. The descending motor commands and ascending sensory information work in conjunction to ensure that the movement is accurate and precise through an error-corrected process that resides in the cerebellum. Disruptions to this information may cause errors in the precision of forelimb motor targeting. According to the previous literature, the external cuneate nucleus (ECN) and central cervical nucleus (CeCv) are responsible for conveying unconscious sensory information from the forelimbs, shoulders, and neck muscles to the cerebellum. Here, we examined the significance of the ECN and CeCv, separately, in forelimb function. In conjunction with inhibitory DREADDs (hM4Di), we observed an obstruction in single pellet reaching and grasping when ECN activity was repressed, both unilaterally and bilaterally, in normal rats. We also observed reduced reach in the grooming assessment bilaterally. We discovered that the CeCv terminates in the medial cerebellar nucleus (MCN), within the deep cerebellar nuclei (DCN), which, to the best of our knowledge, was previously not clearly defined. Together, this information provides evidence that the requirement of ascending sensory information is important in forelimb function.

## 1. Introduction

Goal-directed reaching movements are highly precise and remarkably consistent such that limb trajectory and speed are very similar from trial to trial. It is highly unlikely that this regularity in movement pattern is mediated by feed-forward commands that directly translate descending motor plans into muscle contractions due to the inherent noise in these commands [1,2]. To compensate for noise and ensure regularity, neural feedback strategies provide limb position information to compare and correct limb patterning throughout the entirety of the movement. Relaying sensory information to the cortex for refinement of the planned movement carries inherent temporal delays [3]. One potential mechanism to reduce delay is the copying and conveying of local motor commands to the cerebellum, where they are used to predict sensory consequences of motor action for comparison to the current state of the limb [4]. The corrected motor command is either directly sent to the spinal cord or to brainstem motor centers (e.g., Red nucleus, nucleus giganatocellular) [5,6]. This system is not static but continually adapts to environmental perturbations, muscle fatigue, or changes in the target object dimensions. Adaptation, however, is mediated by the actual movement experienced and not the original motor plan [7]. Therefore, positional information from the limb and cutaneous touch information from the hand and fingers are vital in correction of the motor plan. Surprisingly, little is known about the role of spinocerebellar pathways in mediating forelimb patterning, with the vast majority of studies focusing on nucleus dorsalis of Clark and locomotion [8,9,10]. Proprioceptive information modulates motor control at several different locations, directly within the spinal cord, the cerebellum, and the parietal cortex [11,12]. Genetic deletion of muscle spindles resulted in abnormal hindlimb stride kinematics as well as increased error on the horizontal ladder and uncoordinated limb trajectories during swimming [13,14]. After spinal cord hemisection, spinal proprioceptive activation is required to promote sprouting of spinal circuits to induce recovery of locomotion [13]. The loss of muscle spindles not only prevented the initiation of locomotion but also the maintenance of it, primarily in regions distal to the lesion [14]. Overall, this demonstrates the necessity of sensory information in circuit reorganization leading to locomotor recovery after an incomplete spinal cord injury (SCI). For this study, we examined the involvement of the external cuneate nucleus (ECN) and central cervical nucleus (CeCv) in modulating movement accuracy of the forelimbs. The ECN originates in the caudal medulla, has been shown to be analogous to the dorsal spinocerebellar tract [15,16,17], and is thought to relay unconscious proprioceptive information from the upper limbs and trunk, whereas the CeCv originates in C1–C3 of the spinal cord and has been shown to be analogous to the ventral spinocerebellar tract [16,17], conveying information from neck and shoulder muscle spindles and vestibular system to the cerebellum. Sensory input is thought to refine movement patterns through feedback mechanisms. With the loss of sensory input into the cerebral cortex after cervical SCI, it is necessary to pinpoint another avenue by which sensory feedback can ensure proper functional movement. We hypothesize that the ascending sensory information from the ECN and CeCv into the cerebellum modulates error correction in normal cases and possibly after spinal cord lesion. To examine the significance of these pathways, we utilized intersectional genetics to deliver a modified muscarinic g-protein coupled receptor (hM4Di), which we have shown to effectively silence spinal cord neurons [18]. We observed that the inhibition of the ECN resulted into disruptions in skilled and innate forelimb function in normal animals. We also mapped the termination of the CeCv into the cerebellum, which has not been clearly defined, and found silencing it had little effect on skilled forelimb function. Ultimately, these data will shed light on the association of essential indirect motor pathways with the function and recovery of forelimb patterning and create new studies promoting arm and hand recovery in more severe clinically relevant injury models.

## 2. Materials and Methods

### 2.1. Animals

All surgical and animal care protocols were approved by the Lewis Katz School of Medicine at Temple University’s Institutional Animal Care and Use Committee and performed per the National Institutes of Health Guide for the Care and Use of Laboratory Animals. Male and female Sprague Dawley and Long-Evans rats (40–60 d, 200–350 g; Charles River Laboratories, Horsham, PA, USA) were housed two per cage on a 12 h light dark cycle with food and water provided ad libitum. Female Sprague Dawley rats were used to identify the connections between the ECN and the aIpN. Long-Evans rats we used for behavioral assessment since Long-Evans have been shown to be more favorable for behavioral assessments. Animals were allowed 7 days of acclimatization prior to any experimental procedure. All surgical procedures were performed under aseptic conditions. Animals were anesthetized with ketamine and xylazine, and pain charts were assessed the day of and up to five days after any surgeries performed.

### 2.2. Viral Vectors

Adeno-associated virus serotype 2 (AAV2) was selected as a viral vector to deliver Designer Receptors Exclusively Activated by Designer Drugs (DREADDs). AAV-retro-scCRE and AAV2-syn1-DIO-hM4Di-mCherry (gift from Bryan Roth; Addgene plasmid # 44362) were used to induce inhibition of specific neuronal populations [19], while control animals were injected with AAV2-mCherry. Neuronal labeling studies used AAV2retro-mCherry or AAVretroGFP for retrograde neuronal labeling or AAV1-nucGFP for transsynaptic labeling. All viral constructs utilized the human synapsin (hSyn) promoter and encoded an mCherry fluorescent tag for postmortem histology.

### 2.3. Surgical Procedures

All surgical procedures were carried out under aseptic conditions. To determine the significance of the ECN and CeCv in forelimb function, we injected AAV2 vectors into the cerebellum and cervical spinal cord. Animals were anesthetized using a ketamine/xylazine mixture-administered IP. Animals were then set up in a stereotactic frame (David Kopf Instruments, Tujunga, CA, USA) to position and stabilize the head during injection. A single incision was made with a number 10 scalpel blade in the midline of the head exposing the skull and cranial sutures. Using a motorized drill, windows were opened in various areas of the skull, exposing the anterior interposed nucleus, external cuneate nucleus, or medial cerebellar nuclei. Injections of AAV-retro-CRE were performed either in the anterior IpN [anteroposterior (AP): 11.30 mm, mediolateral (M/L): 2.2 mm, depth (D): 5 mm] or the medial cerebellar nucleus [11.50 mm AP, M/L: 1.0 mm, D: 5.85 mm]. Injections of AAV2-syn1-DIO-hM4Di-mCherry were performed in the external cuneate nucleus [13.1 mm AP, M/L: 2.1 mm, D: 4.3 mm] or the central cervical nucleus (C2-C4) [0.2 mm M/L, D: 1.7 mm]. For spinal cord lesions, C5 laminectomy was performed. Several drops of lidocaine were added to the dura prior to C5 dorsal funiculus lesion. Spring microscissors were marked to a depth of 1.5 mm from dura with an opening of 1.5 mm to sever the dorsal funiculus in between the C4 and C5 dorsal roots. After surgery, animals were administered 0.05 mg/kg buprenorphine (reconstituted in sterile saline) and 3 mL of lactated reader or sterile saline. The day after surgery, animals were given a subcutaneous injection of 0.05 mg/kg buprenorphine twice a day. Animals were given an analgesic (Rimadyl, 1 mg tablet, Cat. No. MD150-2, Bio-Serv, Flemington, NJ, USA) twice a day for 3 additional days thereafter.

### 2.4. Behavioral Training and Assessment

Prior to injections, animals were trained in four separate behavioral assays to establish baselines prior to injury. Single pellet reaching and grasping required animals to be trained 3 times a week for about 4–5 weeks, with some variation in training time between animals as indicated in figure timelines. The variation in training time was dependent on injection times and surgical days to ensure accuracy and precision between animals and cohorts. This did not have any impact on later performance. The training protocol followed was loosely based on [20]. In brief, animals were allowed to become acclimated to the training box apparatus with sucrose pellets on the floor to get the animals used to the taste of the pellets as well as the size and distance of the box. Eventually, sucrose pellets were restricted to just in front of the box to show the only location of the pellets as well as the ability for self-administration. Animals were trained to reach consistently through the slit of the box until a preferred forelimb was established. The animals were encouraged to walk to the back of the box after each successful retrieval based on a weight-dependent indicator, which would not present the pellet fully until the animal walked to the back of the box. Baseline retrieval rates were collected prior to injury using the scoring system outline by [20]. The grooming assay tests how far up the head a rat can reach while grooming. The head is divided into 5 zones, starting at the snout and continuing up the head to above the ears [21]. Animals were sprayed with water or a sucrose-flavored liquid to induce the cleaning cycle and recorded. Recordings were finalized when the animal went through 3 grooming cycles. The Irvine, Beatties, Bresnahan (IBB) scale tests for control of forelimbs and finger dexterity. Animals were put into a glass cylinder and given 3 loop-shaped pieces of cereal and video recorded from beginning to end. Analysis requires watching forelimb and digit control during consumption and giving a numerical score from 0 to 9 based on the criteria met in the IBB protocol [22]. The gridwalk assay evaluates forelimb and hindlimb placements and stepping while walking along a path consisting of ladder rungs. Animals were recorded until 40 forelimb steps were conducted.

### 2.5. Video Recordings

For video recording, a camera (HDR-CX190, Sony or Digital Video Camera, Digital Life, Tokyo, Japan) was placed at an angle conducive to proper featuring of all elements of the behavioral assays conducted. Prior to surgeries, animals were recorded and assessed for baseline behavior. To silence either ECN or CeCv neurons, CNO was used to activate inhibitory Dreadds (hM4Di). Animals were injected with Dreadds prior to the onset of the study, since it takes about 3 weeks for maximal expression. All animals were then recorded and examined on the behavioral assays explained above with and without an intraperitoneal injection of CNO at a dosage of 4 mg/kg on alternating days to induce the DREADDs. All tests involving CNO were started no earlier than 20 min after injection and ceased no later than 60 min after injection to ensure maximal DREADDs activation.

### 2.6. Tissue Processing and Histology

At the conclusion of each experiment, animals were euthanized using 0.5 mL IP injections of pentobarbital (Fatal-Plus, Cat. No. V.P.L. 9373, Vortech, Dearborn, MI, USA). Thoracotomies were performed, and animals were exsanguinated with 0.9% saline before perfusion with 4% paraformaldehyde (PFA) in 0.1 M phosphate buffer (pH 7.5). The brain and spinal cord were carefully dissected out and post-fixed in fresh 4% PFA overnight at 4 °C. The following day, tissue was switched into 30% sucrose in phosphate-buffered saline (PBS) for additional 3 days to cryoprotect tissue. Serially sectioned coronal sections were cut at 30 μm using a cryostat (Superfrost Plus; VWR International) and placed into a cryoprotectant (Na_2_HPO_4_, H_2_O, NaH_2_PO_4_, sucrose, PVP-40, and ethylene glycol), allowing for long-term storage at −20 °C. Slide-fixed sections were washed with phosphate-buffered saline tween (PBS-T) 3 times. To amplify the mCherry signal, sections were incubated with primary antibody dsRed (rabbit, polyclonal; Cat. No. 632496, Takara Bio Inc., Mountain View, CA, USA) at 1:500 overnight at 4 °C. Sections were then washed in PBS-T 3 times and incubated with secondary antibody Alexa Fluor 594 (donkey anti-rabbit; Cat. No. 11-585-144, Jackson Immunoresearch Laboratories Inc., West Grove, PA, USA) and/or Alexa Fluor 488 (donkey anti-goat; Cat. No. 705-545-147, Jackson Jackson Immunoresearch Laboratories Inc., West Grove, PA, USA) at 1:500 for 2 h at room temperature. After washing with PBS-T 3 times, sections were warmed on a heater (XH-2002 Slides Warmer, Premiere Instruments, Manassas, VA, USA) and cover-slipped with Fluoromount-G (Cat. No. 0100-01, VWR International, Radnor, PA, USA). Images were acquired using a Zeiss microscope (Jena, Germany) at 5× and 10× magnitude. The 5× magnitude images were stitched together using Adobe Photoshop. To determine the percent of labeled neurons, mCherry- or GFP- positive neurons were counted and compared to the total number of neurons labeled using the NeuN antibody (Cat No. MAB377, Millipore/Sigma, Burlington, NJ, USA) within either the ECN or CeCv. The Brain Atlas [23] was used to verify the location of the ECN, and the CeCv and counts were performed manually.

### 2.7. Statistics

Behavioral assays were analyzed according to the region of interest that each assay was used for as stated earlier. Each test was measured with and without CNO administration, pre- and post-lesioned. The single pellet reaching and grasping assay was measured based the number of misses compared to the number of successful reaches out of a total of twenty pellets or twenty minutes from the beginning the test. The gridwalk test accounted for differences in the number of missed and successful steps taken per trial, which was about forty steps. The IBB test measured the differences in forelimb control and finger dexterity. The grooming test measured the differences in grooming cycles between the total amount of grooming cycles in a trial. All assays were analyzed, in a blinded fashion, and executed using one-way ANOVA and Tukey multiple comparisons via Graphpad Prism. A *p*-value < 0.05 was considered significant. Data are presented as mean ± standard error of the mean (SEM).

## 3. Results—Sensory Input Is Involved in Forelimb Function

Coordinating motor movements requires sensory information to accurately guide and correct potential errors that might arise from the movement. This is mostly organized by conscious somatosensory information extending into the cortex. However, unconscious proprioceptive information via the spinocerebellar pathways might act as a supplemental pathway to aid coordination in the absence of conscious somatosensory pathways. Previous research indicates the external cuneate nucleus (ECN) and the central cervical nucleus (CeCv) represent the spinocerebellar pathways for the forelimbs. These nuclei were individually silenced in either normal rats or those receiving C5 dorsal column lesions. To silence each of these nuclei, we used intersectional genetics to target the expression of DREADDs to the respective nucleus.

To validate targeting of ECN neurons, we injected AAV2-retro-GFP into the anterior interposed nucleus (aIpN) and AAV2-mCherry into the ECN (Figure 1A,B) to produce double-labeled nuclei in the ECN. These injections resulted in green-labeled neurons in the aIpN (Figure 1C), as well as neurons projecting from the ECN into the alpN (Figure 1D,G). ECN neurons were also labeled red from the injection of AAV2-mCherry directly into the ECN (Figure 1E,H), confirming that we pinpointed the coordinates correctly. The majority of ECN neurons were double-labeled (yellow-labeled ECN neurons), confirming the connection between the aIpN and ECN (Figure 1I,F). We performed analyses to determine the percentage of ECN neurons projecting into the aIpN, ECN neurons, and double-labeled neurons in the ECN compared to the total number of ECN neurons in the caudal medulla of the brainstem (Figure 1J). We observed similar labeling of ECN neurons between each rat within this cohort, showing an average of 80%, 90%, and 63% of the total ECN neurons labeled with either GFP, mCherry, or both, respectively. These findings show the consistency between injections within the rats as well as provide evidence on the relationship between aIpN and the ECN.

A total of 13 rats from two separate cohorts were used to examine changes in behavioral responses after silencing of the ECN. All rats were first trained to perform single pellet reaching studies three times a week for 4 weeks (Figure 2A). After training, we utilized the two-viral vector approach using retrograde CRE-expressing adeno-associated virus (retroAAV-scCRE) and an CRE-dependent adeno-associated virus containing inhibitory DREADDs (AAV2-hSyn-DIO-hM4Di-mCherry) with a human synapsin promoter to drive transgene expression (Figure 2B). Following the collection of baseline behavioral training, injections were conducted in Long Evans uninjured rats (Figure 2A,B). We, unilaterally, injected retro-AAV-scCRE into the aIpN and AAV2-hSyn-DIO-hM4Di-mCherry or AAV2-mCherry into the ECN to distinguish between experimental and control, respectively (Figure 2B). Animal groups were given 2 weeks for recovery, followed by 2 -4 weeks of retraining to be brought back to baseline levels in all the behavioral assessments. To silence neurons, intraperitoneal injections of clozapine-N-oxide (CNO, 4 mg/kg) or vehicle (control) were given on alternating days to all rats for all behavioral assessments (Figure 2C–F). Using single pellet retrieval assays, we observed that silencing the ECN resulted in statistically significant differences and a 15% increase in reaching and grasping errors when compared to control groups for that assay (Figure 2C). The grooming test measures how far towards the top and back of the head a rat can reach while grooming. To determine if silencing the ECN negatively affected reaching during grooming, we delivered CNO or vehicle on alternating days to 15 normal rats from 2 separate cohorts of rats (Figure 2E). Again, we observed that silencing ECN neurons resulted in a statistically significant reduction in reaching while grooming when compared to all other control groups (Figure 2E). To examine if silencing ECN neurons negatively affect hindlimb placement on a horizontal ladder, we assayed the number of hindlimb falls though the rungs of the ladder. There was no difference between any of the groups (Figure 2D). We also observed no negative affect of silencing ECN neurons on forelimb and digit dexterity while manipulating food during eating using the Irvine, Beatties and Bresnahan assay (IBB, Figure 2F). We did not repeat these studies after dorsal column injuries because the DRG proprioceptive sensory axons travel through the dorsal columns to the ECN [24], so a post-lesion cohort would be deemed unnecessary. Immunohistochemistry (IHC) was performed to confirm injection sites and the percentage of neurons labeled (Figure 2G,H). These findings show that the sensory input from the ECN participates in mediating forelimb reaching but not foot placement during gridwalk or wrist movements and digit dexterity.

As stated before, the cerebellum receives sensory input from the ECN and the CeCv for real-time error correction. Previous research has pointed to the anterior interposed nucleus (aIpN) and posterior interposed nucleus (pIpN) as potential locations for the termination of the CeCv. Therefore, in separate cohorts, we injected AAV2-retro-mCherry into the aIpN and the pIpN to assess labeling of red neurons in the CeCv (Figure 3B,C). Although we observed some red neurons in the CeCv from the aIpN and the pIpN (Figure 3E,F), the labeling was not consistent amongst all sections, indicating another nuclei for termination of the CeCv axons. Therefore, we, bilaterally, injected an AAV1-GFP virus into the area of the CeCv to determine the location of these cerebellum neurons (Figure 4A,B). This serotype of AAV provides the ability for the virus to travel trans-synaptically into the postsynaptic neuron. This injection labeled neurons within the medial cerebellar nuclei (MCN, or fastigial nuclei) (Figure 4C). To further confirm this discovery, we injected AAV2-retro-mCherry into the MCN to see if neurons would be detected within the CeCv (Figure 4B). After IHC and analysis, we observed neurons specifically within the CeCv (Figure 4D,E). Here, we identified 4–6 neurons per tissue section, accounting for animal size differences. It was similar to that observed in cats [8–10 neurons/section, [25]. This finding confirms the discovery of the central cervical nucleus terminations within the medial cerebellar nuclei. Upon the appropriate termination destination of the CeCv, we utilized the same two-way viral vector approach as discussed with the ECN cohort (Figure 5A,B). Unfortunately, in pre-lesioned and post-lesioned animals, we were unable to achieve statistical significance in the behavioral assessments (Figure 5C,D).

## 4. Discussion

Spinal cord injury dramatically affects motor function by disrupting descending motor and ascending sensory pathways. These pathways work in tandem to ensure effective processing and actual movement. Previously, we observed that after a dorsal funicular (DF) lesion at the cervical level (C5), rats recovered single pellet reaching but not digit and wrist dexterity with rehabilitation [18]. This lesion not only cuts the dorsal column sensory and corticospinal pathways but also sensory afferents extending into the ECN, which travel through the dorsal columns [24]. Since these rats recover almost to baseline levels with rehabilitation after C5 DF lesions, we were curious to know if silencing the ECN in non-injured rats participated in the accuracy for reaching and grasping food pellets. Here, we found that silencing of the external cuneate nucleus (ECN), located in the caudal medulla, reduced the accuracy of forelimb targeting during the single pellet reaching assay as well as reaching in grooming assays. However, silencing the ECN showed no disruption of digit and wrist movements indicating grasping. This might be due to the fact that both the dorsal columns sensory and corticospinal tract remained intact and that grasping does not require direct proprioceptive feedback through the cerebellum. Previous studies show that lesions to both the aIpN and pIpN did not impair grasping but decreased the accuracy of reaching movements [26]. Since dorsal funiculus lesions would cut the sensory afferent entering the ECN, we did not repeat this study in rats with spinal cord injury.

In both monkeys and rats, the ECN has two efferent pathways—one extending into the cerebellum and the other extending into the ventral posterior thalamic nucleus (VPN) [27,28]. For these studies, we chose to silence the cerebellar pathway and not the thalamic pathway since it seems to be part of the conscious proprioception to the cortex. To target the ECN/cerebellar neurons, we first established that the ECN terminates in the anterior interposed nuclei [29,30]. Using a Cre-dependent two-way viral vector approach, we were able to inhibit the cerebellar projections from the ECN, both unilaterally and bilaterally. Even though we first determined which forepaw they preferred to reach with, they would often switch paws when silenced by CNO and needed to be removed from the study. We then switched to bilateral silencing with the other cohorts to reduce overall animal numbers needed. Bilateral lesions showed more consistent reaching deficits even when they switched forepaws.

Part of the reaching maneuver requires neck muscles to turn the head toward the target and shoulder muscles to aid in lifting the arm to reach through the slit in the plexiglass wall. The central cervical nucleus (CeCv) was shown to relay proprioceptive information from the neck and shoulder muscles and vestibular information for neck rotation to the cerebellum [31,32]. It is also thought to be analogous to the ventral spinocerebellar tract from the hindlimbs [16,17]. However, the previous literature identified that CeCv synaptic terminates within multiple deep cerebellar nuclei [33]. In our study, we showed that the CeCv terminates within the medial cerebellar nuclei (MCN) or fastigial nucleus. Originally, we thought the CeCv could supplement proprioceptive information lost after dorsal column lesions to assist with the reaching response. Even though we observed good labeling of the CeCv using our two-way viral vector approach, we were unable to achieve statistical significance with any of the behavioral assessments, either pre- or post-lesioned. This could simply be due to the proprioception of the neck and shoulders not being important in mediating reaching movements. However, we observed good recovery of forelimb targeting with 4 weeks of rehabilitation after dorsal funiculus lesions [18], which did not seem to be dependent on proprioception from either the CeCv or the ECN. We hypothesized that inhibiting the CeCv would cause a major disruption in behavioral movements because after an SCI, the CeCv remains intact while much of the ascending sensory information being sent to the cortex or cerebellum is lesioned.

We believe that this is due to a couple of factors. We hypothesize that there are secondary pathways that relay proprioceptive information to the cerebellum independent of the ECN or CeCv. The CeCv is only within the upper cervical spinal cord (C1–C3) and may primarily involve muscles innervated by cranial nerve XI and VIII, since it is known to receive sensory information from these muscles and some vestibular information for neck rotation. However, most of the spinal motor neurons for arm, wrist, and digit musculature required for reaching and grasping are localized to the C3–T1 spinal cord. Interestingly, sensory and motor information from these regions is relayed to the lateral reticular nucleus (LRN) via either direct innervation or through the C3/C4 propriospinal neurons [1,34]. The LRN in turn relays directly to the cerebellum, primarily the vermis and para vermal regions [35]. The LRN is also active during the reaching movement and, when perturbed, shows reaching deficits [1]. Additionally, another alternative pathway could be mediated through the proprioceptive 1a afferents within the dennervated spinal cord distal the lesion. Takeoka and Arber show that acute ablation of muscle spindles prior to spinal cord hemisection showed major deficits in locomotion; however, muscle spindle ablation after recovery from SCI showed significantly less deficits in locomotion, most likely by supporting spinal plasticity prior to removal of proprioceptive input [14]. Our procedure silenced the ECN and CeCv after spinal cord injury and rehabilitation, potentially showing better recovery than if we silenced these neurons during the recovery period.

Overall, we have highlighted the importance of sensory input in movement. These data will shed light on the association of essential indirect motor pathways with the function and recovery of skilled forelimb patterning and provide an avenue to new studies promoting arm and hand recovery in more severe clinically relevant injury models.

## Figures and Tables

**Figure 1 cells-14-00589-f001:**
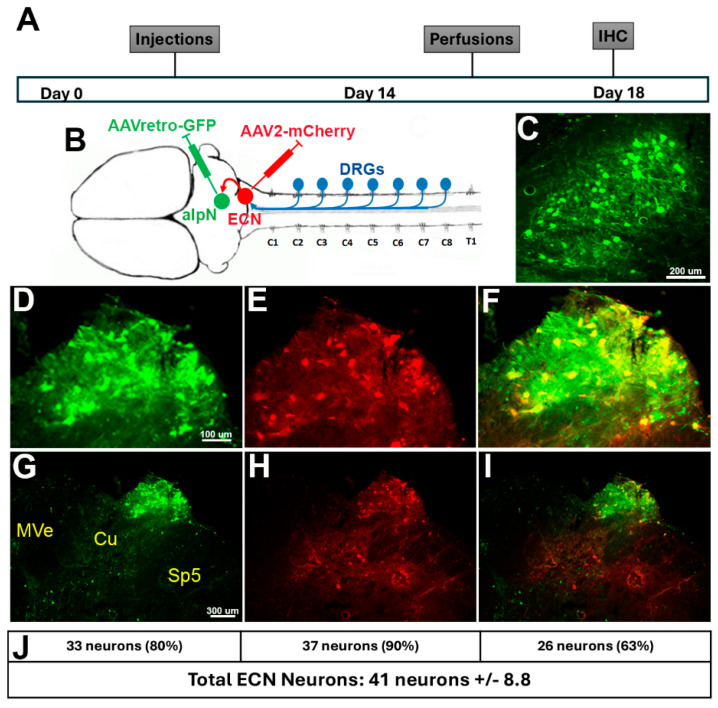
Confirmation of the external cuneate nucleus (ECN) projecting to the anterior interposed nucleus (aIpN) using adeno-associated viruses (AAVs). (**A**) Timeline from viral injections into the cerebellum and brainstem to the immunohistochemistry (IHC). Typically, the process averages ~18 days. (**B**) Illustration showing the circuitry, with the DRGs (blue) sending collaterals into the ECN (red) and the ECN connection to the aIpN (green). We unilaterally injected AAV2-retro-GFP into the anterior interposed nucleus (aIpN), which not only labels the aIpN (**C**) but also retrogradely labels neurons within the external cuneate nucleus (**D**,**G**) Verification that ECN was targeted with directly injected AAV2-mCherry into the ECN (**E**,**H**). (**F**,**I**) Double labeling of the ECN, validating the connection between the aIpN and the ECN, showing most are double-labeled yellow neurons. Medial vestibular nucleus (MVe), cuneate nucleus (Cu), and spinal nucleus of cranial nerve V (Sp5). (**J**). The mean neuronal counts and percentages of labeled neurons within ECN as indicated by the number of neurons per section.

**Figure 2 cells-14-00589-f002:**
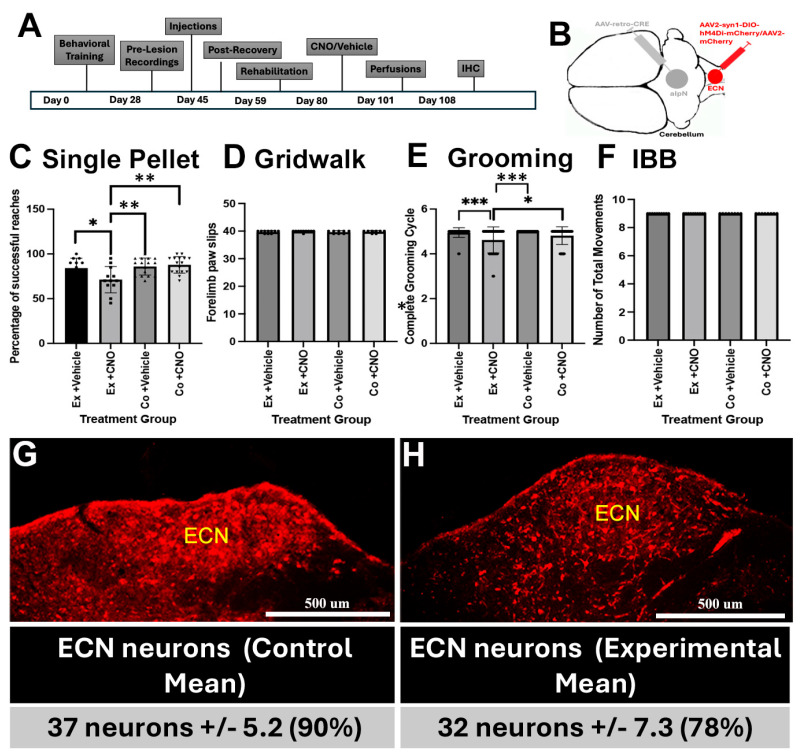
Inhibition of the ECN disrupts forelimb function in pre-lesioned rats. (**A**) Timeline showing scheduling of all procedures in the study. (**B**) Illustration showing injections of AAV2-retro-CRE into the anterior interposed nucleus (aIpN) and AAV2-syn1-DIO-hM4Di-mCherry to silence neurons or AAV2-mCherry (control) into the ECN. (**C**) Combined analyses of behavioral testing of two separate cohorts. Single pellet reaching scores after administration of CNO (4mg/kg) or vehicle on alternating days showed a 16% decrease between the experimental and control groups. Experimental vehicle (●) vs. experimental CNO (■), * *p* = 0.0299, experimental CNO vs. control vehicle (▲), ** *p* = 0.0055, experimental CNO vs. control CNO (▼), ** *p* = 0.0014. (**D**) Behavioral testing of the gridwalk assessment. (**E**) Analysis of reaching during grooming showed a 14% decrease between the experimental and control groups. Experimental vehicle vs. experimental CNO, *** *p* = 0.0002, experimental CNO vs. control vehicle, *** *p* ≤ 0.0001, experimental CNO vs. control CNO, * *p* = 0.05. (**F**) Behavioral testing of the IBB assessment. Immunohistochemistry (IHC) showing neuronal expression within control (**G**) or hM4Di experimental (**H**) groups. Neuronal count and percentages of the experimental ECN were assessed as in Figure 1. Statistical analysis was performed with an ordinary one-way ANOVA with a Tukey’s comparison test. Data are represented as means ± SEMs.

**Figure 3 cells-14-00589-f003:**
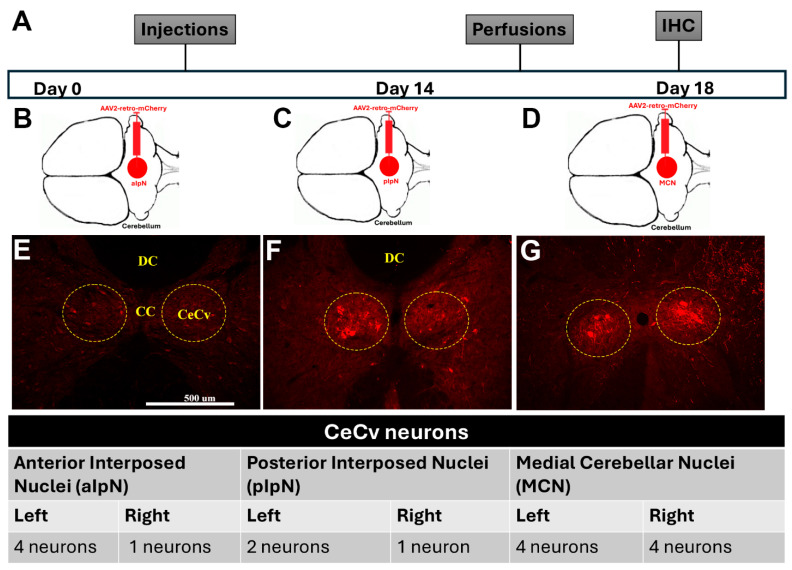
The CeCv innervates the posterior interposed nucleus (plpN) and medial cerebellar nucleus (MCN). (**A**) Timeline for scheduling of procedures used in the study. (**B**) Illustration showing injection site for AAV2-retro-mCherry into the aIpN. (**C**) Illustration showing the injection site for AAV2-retro-mCherry into the pIpN. (**D**) Illustration showing the injection site for AAV2-retro-mCherry into the MCN. (**E**) Immunohistochemistry (IHC) displaying red neurons in the CeCv (yellow circles) projecting from the aIpN (**E**), plpN (**F**), or MCN (**G**). Mean neuronal counts of the CeCv were evaluated per section.

**Figure 4 cells-14-00589-f004:**
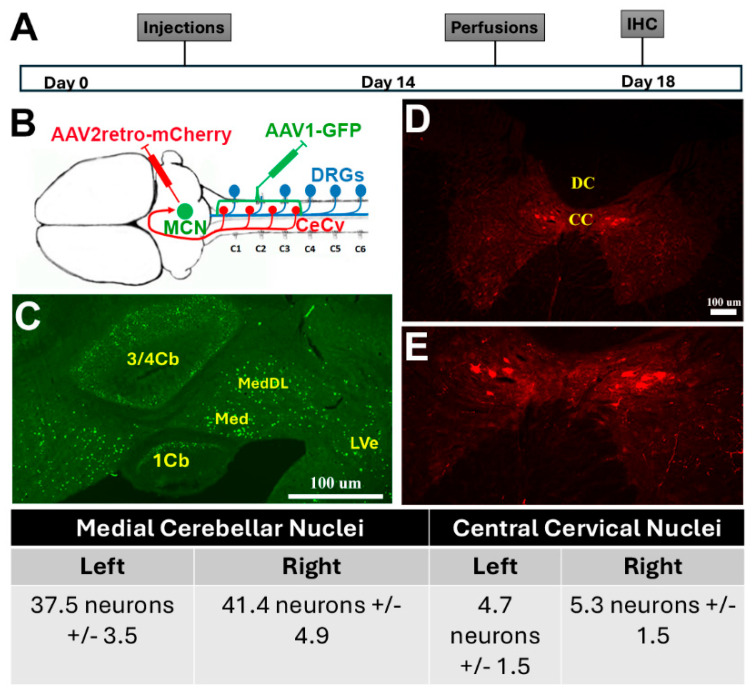
Identifying projection from the central cervical nucleus (CeCv) into the medial cerebellar nucleus (MCN). (**A**) Timeline showing scheduling of procedures for the study. (**B**) Illustration showing the circuit from DRG neurons to CeCv to MCN. To identify the target of the CeCv neurons, we injected AAV1-scGFP into the C2–C4 spinal cord to trans-synaptically label neurons within the deep cerebellar nuclei. (**C**) Cervical injections indicated the postsynaptic targets to be within the MCN. (**D**,**E**) High and low magnifications showing mCherry-labeled CeCv neurons within the spinal cord after injection of AAVretro-mCherry into the MCN. Neuronal counts of the MCN were evaluated per section. Medial cerebellar nucleus (Med), dorsal/lateral Med (MedDL), lateral vestibular nucleus (LVe), cerebellar regions (1Cb; 3/4Cb).

**Figure 5 cells-14-00589-f005:**
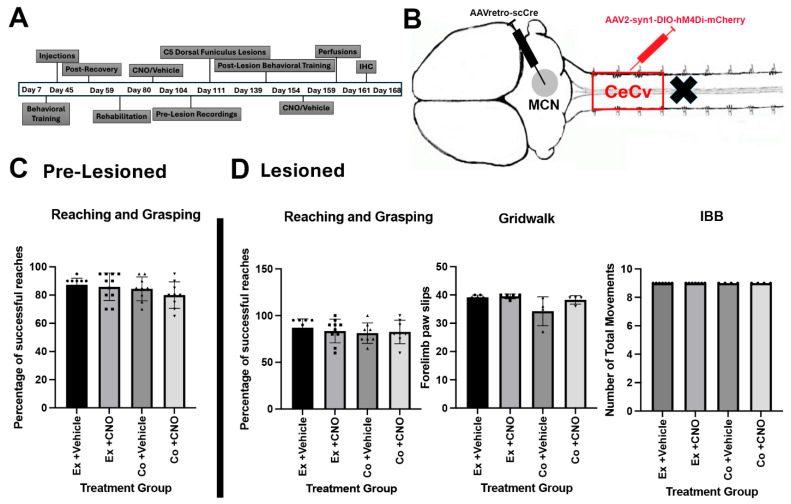
Silencing CeCv neurons projecting to the MCN does not disrupt forelimb function in normal or lesioned rats. (**A**) Timeline of procedures used in this study. (**B**) Illustration showing targeting of expression to the CeCv by injecting AAV2-retro-CRE into the MCN and either AAV2-syn1-DIO-hM4Di-mCherry or AAV2-mCherry into the C2–C4 cervical spinal cord. (**C**) Pre-lesion analysis of single pellet reaching assay showing no statistical significance between experimental (ex) and control (Co) groups with or without CNO treatment. (**D**) Post-lesioned behavioral assessments of single pellet reaching, gridwalk, or digit and wrist dexterity (IBB) showed no differences. Statistical analysis was performed using one-way ANOVA with Tukey’s comparison test. Data are represented as means ± SEMs. Symbols are same as those in Figure 2 and represent individual animals test.

## Data Availability

The original contributions presented in this study are included in the article. Data will be made available to anyone upon request to the corresponding author.

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
