# Peer review of "The Sensory Input from the External Cuneate Nucleus and Central Cervical Nucleus to the Cerebellum Refines Forelimb Movements"

_cells, 2025, doi:10.3390/cells14080589_

Round 1
Reviewer 1 Report
Comments and Suggestions for Authors
This manuscript studied the effect of forelimb movement by inhibition of neurons in the ECN and CeCv. Using DREADDs techniques to inhibit the activity of ECN neurons in rats, authors observed an obstruction in forelimb reaching and grasping and grooming tests. They discovered that the CeCv terminates in the medial cerebellar nucleus rather than the interposed nucleus within the deep cerebellar nuclei (DCN), which was previously not clearly defined. The results provide evidence that the ascending sensory information is required to modulate forelimb function. The discovery is very interesting and important. The experimental design is appropriate and the technique is advanced. The findings will enrich the current knowledge of motor-sensory control. The writing is clear and the whole text is easy to read. However, there are several missing parts in the methods and some minor concerns that need to be addressed to improve the manuscript before publication.
- It would help understand the neuronal pathways if providing a cartoon outline to summarize the connection disclosed in this study between the ECN/CeCv and different nuclei in the cerebellum.
- In line 110 on page 4, delete one “the”.
- It didn’t seem correct for the dose of 0.5 g of buprenorphine per rat.
- In the method part, it lacked a description of how the nuclei of ECN or CeCv were recognized, and the total number of these nuclei was counted.
- Describe how CNO was used in the method.
- There was no information about how the dorsal collum lesion at the C5 spinal level was performed.
- In Fig. 1, the values of labeled neuron percentage, 80, 90, 63% are not matched with those reported in the text (line 211). Are these for only one rat? If so, this should be indicated in the legend.
- In line 238 on page 6, it appeared to miss words. Should it be “injected retro-AAV-scCRE into the aIpN and AAV2-hSyn-DIO-hM4Di-mCherry or 238 AAV2-mCherry into the ECN”?
- In the grid walk test in line 252 on page 6, it was inappropriate to describe a rodent with “foot” or hand. Please replace it with “hindlimb”.
- Since this study includes a very small portion of C5 dorsal funiculus lesion in which the data is negative, and the ECN-Cerebellum pathway is not a spinocerebellar tract, the title of this manuscript may need to be revised, e.g., “The sensory input from the ECN and CeCv to the cerebellum refines forelimb movements” or others.
Author Response
We thank the reviewer for their advice and corrected the manuscript to accordingly.
- It would help understand the neuronal pathways if providing a cartoon outline to summarize the connection disclosed in this study between the ECN/CeCv and different nuclei in the cerebellum.
- In line 110 on page 4, delete one “the”.
Response: Done (now line 118)
- It didn’t seem correct for the dose of 0.5 g of buprenorphine per rat.
Response: Correct amount added 0.05mg/kg
- In the method part, it lacked a description of how the nuclei of ECN or CeCv were recognized, and the total number of these nuclei was counted.
Response: To better explain this labeling we added information bout both the viruses and quantitative methods sections. The neurons were simply identified for there fluorescent expression.
Added to viral vector section: Neuronal labelling studies used AAV2retro-mCherry or AAVretro-GFP for retrograde neuronal labeling or AAV1-nucGFP for transsynaptic labeling of postsynaptic neuron nuclei.
Added to the Histology section: To determine the percent of labelled neurons, mCherry- or GFP- positive neurons were counted and compared to the total number of neurons labeled using NeuN antibody (Cat No. MAB377, Millipore/Sigma) within either the ECN or CeCv. The Brain Atlas (Paxinos and Watson, 2014) was used to verify the location of the ECN and the CeCv and counts were done, manually.
- Describe how CNO was used in the method.
Response: In Video recording section: To silence either ECN or CeCv neurons, CNO was used to activate inhibitory Dreadds (hM4Di). Animals were injected with Dreadds prior to the onset of the study, since it takes about 3 weeks for maximal expression. All animals were then recorded and examined on the behavioral assays explained above with and without an intraperitoneal injection of CNO at a dosage of 4 mg/kg on alternating days, to induce the DREADDs. All tests involving CNO were started no earlier than 20 minutes after injection and ceased no later than 60 minutes after injection to ensure maximal DREADDs activation
- There was no information about how the dorsal collum lesion at the C5 spinal level was performed.
Response: For spinal cord lesions, C5 laminectomy was performed. Several drops of lidocaine were added to the dura prior to C5 dorsal funiculus lesion. Spring microscissors were marked to a depth of 1.5 mm from dura with an opening of 1.5 mm to sever the dorsal funiculus in between the C4 and C5 dorsal roots.
- In Fig. 1, the values of labeled neuron percentage, 80, 90, 63% are not matched with those reported in the text (line 211). Are these for only one rat? If so, this should be indicated in the legend.
Response: The counts from the figure are correct and changed in the text to match the figure.
- In line 238 on page 6, it appeared to miss words. Should it be “injected retro-AAV-scCRE into the aIpN and AAV2-hSyn-DIO-hM4Di-mCherry or 238 AAV2-mCherry into the ECN”?
Response: Corrected: We, unilaterally, injected retro-AAV-scCRE into the aIpN and AAV2-hSyn-DIO-hM4Di-mCherry or AAV2-mCherry into the ECN to distinguish between experimental and control, respectively (Fig. 2B).
- In the grid walk test in line 252 on page 6, it was inappropriate to describe a rodent with “foot” or hand. Please replace it with “hindlimb”.
Response: Done.
- Since this study includes a very small portion of C5 dorsal funiculus lesion in which the data is negative, and the ECN-Cerebellum pathway is not a spinocerebellar tract, the title of this manuscript may need to be revised, e.g., “The sensory input from the ECN and CeCv to the cerebellum refines forelimb movements” or others.
Response: Sounds good.
Reviewer 2 Report
Comments and Suggestions for Authors
The paper provides new experimental insights into the role of proprioceptive feedback in forelimb reaching. The neural systems selected and how they were experimentally manipulated were reasonable as were the behavioral tests used. I have no major concerns with the experimental design and analyses. There are some stylistic changes needed, and several issues need clarification and elaboration. How the two strains of rats were distributed across experiments could be a major issue if only one strain were used per procedure, as this could greatly impact the ability to compare and contrast the role of the two brain regions studied (detailed in my comments below). If the two strains were evenly distributed, then a simple clarification in the Methods would suffice, but if they were not, then a more thorough re-evaluation of what can be concluded from the findings is needed in the Discussion. Otherwise, most of the comments I make should be easily addressed.
Line 48: Change “...modulates motor control at several different location...” to “...modulates motor control at several different locations...”
Line 49: Change “...and parietal cortex” to “...and the parietal cortex”
Line 51: Change “...in the abnormal hindlimb...” to “...in abnormal hindlimb...”
Line 58: As this is the first time SCI is mentioned, what the abbreviation stands for should be stated.
Line 68: Chane “We hypothesis...” to “We hypothesize...”
Lines 69-70: The phrase “...modulate error correctness in normal and possibly after spinal cord lesions” is not clear. First by ‘correctness’ do you mean ‘correction’ because the systems correct errors, they do not ensure that the errors are correct. Second, it is not clear how these circuits can correct movements in both normal and transected animals, would it not be the capacity lost in the transected animals? Please correct or clarify.
Line 76: Change “...this data...” to “...these data...” (datum is singular, data is plural)
Line 85: Why were SD and LE rats used? While both strains have similar success rates in obtaining pellets in the single pellet reaching task, the SD rats exhibit greater motoric deviation in many of the subcomponent movements involved (see Whishaw, I. Q., Gorny, B., Foroud, A. & Kleim, J. A. (2003). Long–Evans and Sprague–Dawley rats have similar skilled reaching success and limb representations in motor cortex but different movements: some cautionary insights into the selection of rat strains for neurobiological motor research. Behavioural Brain Research, 145, 221-232). A rationale is needed for why these two strains were used, and whether how they were used and assessed took these movement differences into account.
Lines 121-122: Variation in training time among the animals is mentioned but not explained. What was the variation and what criteria were used to warrant the variation? Did this have an impact on later performance?
Lines 135-136: Change “...when the animal goes through...” to “...when the animal went through...” Also, it is not made clear what was measured and how the grooming assesses paw use. Finally, while references are provided for the other behavioral tests, one is not included for grooming. For readers unfamiliar with rat grooming behavior, it would be useful to provide a suitable reference.
Lines 230-298: Thirteen rats from 2 separate cohorts are mentioned (line 230) and specific reference is made to LE rats for one procedure (line 237) but is not made explicit if both SD and LE rats were used in a counter-balanced manner for all procedures and tests. Given the strain differences in motoric skills between these two strains (see above comment), if one or other strain is used for any one procedure and the opposite strain for the other procedure, then any differences may not be due to the neural system that is compromised, but rather, to the strain of rat used. Please specify how the LE and SD rats were distributed across the experimental manipulations.
Line 246: Change “The grooming test measure how far towards...” to “The grooming test measures how far towards...”
Line 358: Change “For these studies we choose silence...” to “For these studies, we choose to silence...”
Line 365: Change “We then switch to...” to “We then switched to...”
Line 375: I suspect that “...previous literature identified CeCv synaptic terminates within...” should be “...previous literature identified that CeCv synapses terminate within...”
Lines 381-385: Not clear why ‘however’ is used to link these two sentences. I do not understand what is being contrasted, and so what needs to be reconciled. This confusion impacts the hypothesis posited in the next paragraph. Please restate and clarify.
Line 388: Change “...only is...” to “...is only...”
Author Response
We thank the reviewer for their helpful comments and corrected the manuscript to accordingly.
Line 48: Change “...modulates motor control at several different location...” to “...modulates motor control at several different locations...”
Response: Done
Line 49: Change “...and parietal cortex” to “...and the parietal cortex”
Response: Done
Line 51: Change “...in the abnormal hindlimb...” to “...in abnormal hindlimb...”
Response: Done
Line 58: As this is the first time SCI is mentioned, what the abbreviation stands for should be stated. Response: Done
Line 68: Chane “We hypothesis...” to “We hypothesize...”
Response: Done
Lines 69-70: The phrase “...modulate error correctness in normal and possibly after spinal cord lesions” is not clear. First by ‘correctness’ do you mean ‘correction’ because the systems correct errors, they do not ensure that the errors are correct. Second, it is not clear how these circuits can correct movements in both normal and transected animals, would it not be the capacity lost in the transected animals? Please correct or clarify.
Response: Rewritten: Sensory input is thought to refine movement patterns through feedback mechanisms. With the loss of sensory input into the cerebral cortex after cervical SCI, it is necessary to pinpoint another avenue by which sensory feedback can ensure proper functional movement. We hypothesize that the ascending sensory information from the ECN and CeCv into the cerebellum modulates error correction in normal and possibly after spinal cord lesion.
Line 76: Change “...this data...” to “...these data...” (datum is singular, data is plural)
Response: Done
Line 85: Why were SD and LE rats used? While both strains have similar success rates in obtaining pellets in the single pellet reaching task, the SD rats exhibit greater motoric deviation in many of the subcomponent movements involved (see Whishaw, I. Q., Gorny, B., Foroud, A. & Kleim, J. A. (2003). Long–Evans and Sprague–Dawley rats have similar skilled reaching success and limb representations in motor cortex but different movements: some cautionary insights into the selection of rat strains for neurobiological motor research. Behavioural Brain Research, 145, 221-232). A rationale is needed for why these two strains were used, and whether how they were used and assessed took these movement differences into account.
Response: We started with SD rats for tracing, but found LE rats to be easier to train for behavioral studies. We rewrote this bit for clarity. “Female Sprague Dawley rats were used to identify the connections between the ECN and the aIpN. Long-Evans rats we used for behavioral assessment since Long-Evans have been shown to be more favorable for behavioral assessments.
Lines 121-122: Variation in training time among the animals is mentioned but not explained. What was the variation and what criteria were used to warrant the variation? Did this have an impact on later performance?
Response: Rewritten to read: Single pellet reaching and grasping required animals to be trained 3 times a week for about 4-5 weeks, with some variation in training time between animals as indicated in figure timelines. The variation in training time was dependent on injection times and surgical days to ensure accuracy and precision between animals and cohorts. This did not have any impact on later performance.
Lines 135-136: Change “...when the animal goes through...” to “...when the animal went through...” Also, it is not made clear what was measured and how the grooming assesses paw use. Finally, while references are provided for the other behavioral tests, one is not included for grooming. For readers unfamiliar with rat grooming behavior, it would be useful to provide a suitable reference.
Response: Rewritten: The grooming assay tests how far up the head a rat can reach while grooming. The head is divided into 5 zones, starting at the snout and continuing up the head to above the ears (Gensel et al., 2006).
Lines 230-298: Thirteen rats from 2 separate cohorts are mentioned (line 230) and specific reference is made to LE rats for one procedure (line 237) but is not made explicit if both SD and LE rats were used in a counter-balanced manner for all procedures and tests. Given the strain differences in motoric skills between these two strains (see above comment), if one or other strain is used for any one procedure and the opposite strain for the other procedure, then any differences may not be due to the neural system that is compromised, but rather, to the strain of rat used. Please specify how the LE and SD rats were distributed across the experimental manipulations.
Response: As indicated in the earlier comment, SD rats were only used in the anatomical assessment and not behavioral studies. Sorry for the confusion.
Line 246: Change “The grooming test measure how far towards...” to “The grooming test measures how far towards...” Response: Corrected
Line 358: Change “For these studies we choose silence...” to “For these studies, we choose to silence...” Response: Corrected
Line 365: Change “We then switch to...” to “We then switched to...” Response: Corrected
Line 375: I suspect that “...previous literature identified CeCv synaptic terminates within...” should be “...previous literature identified that CeCv synapses terminate within...” Response: Corrected
Lines 381-385: Not clear why ‘however’ is used to link these two sentences. I do not understand what is being contrasted, and so what needs to be reconciled. This confusion impacts the hypothesis posited in the next paragraph. Please restate and clarify.
Response: I hope this clarifies your confusion: “We hypothesized that inhibiting the CeCv would cause a major disruption in behavioral movements, because after a SCI, the CeCv remains intact while much of the ascending sensory information being sent to the cortex or cerebellum would be lesioned.”
Line 388: Change “...only is...” to “...is only...” Response: corrected